# Roles of *AGD2a* in Plant Development and Microbial Interactions of *Lotus japonicus*

**DOI:** 10.3390/ijms23126863

**Published:** 2022-06-20

**Authors:** Mingchao Huang, Mengru Yuan, Chunyu Sun, Meiru Li, Pingzhi Wu, Huawu Jiang, Guojiang Wu, Yaping Chen

**Affiliations:** 1Key Laboratory of South China Agricultural Plant Molecular Analysis and Genetic Improvement & Guangdong Provincial Key Laboratory of Applied Botany, South China Botanical Garden, Chinese Academy of Sciences, Guangzhou 510650, China; huangmc08@163.com (M.H.); yuanmengru@scbg.ac.cn (M.Y.); emetselch2022@126.com (C.S.); limr@scbg.ac.cn (M.L.); hwjiang@scbg.ac.cn (H.J.); wugj@scbg.ac.cn (G.W.); 2University of Chinese Academy of Sciences, Beijing 100049, China; 3Key Laboratory of South Subtropical Fruit Biology and Genetic Resource Utilization, Ministry of Agriculture/Key Laboratory of Tropical and Subtropical Fruit Tree Research of Guangdong Province, Institution of Fruit Tree Research, Guangdong Academy of Agricultural Sciences, Guangzhou 510640, China; pzwu@scbg.ac.cn

**Keywords:** *Lotus japonicus*, development, nodulation, disease resistance

## Abstract

Arabidopsis AGD2 (Aberrant Growth and Death2) and its close homolog ALD1 (AGD2-like defense response protein 1) have divergent roles in plant defense. We previously reported that modulation of salicylic acid (SA) contents by *ALD1* affects numbers of nodules produced by *Lotus japonicus*, but *AGD2′*s role in leguminous plants remains unclear. A combination of enzymatic analysis and biological characterization of genetic materials was used to study the function of *AGD2* (*LjAGD2a* and *LjAGD2b*) in *L. japonicus*. Both *LjAGD2a* and *LjAGD2b* could complement *dapD* and *dapE* mutants of *Escherichia coli* and had aminotransferase activity in vitro. *ljagd2* plants, with insertional mutations of *LjAGD2*, had delayed flowering times and reduced seed weights. In contrast, overexpression of *LjAGD2a* in *L. japonicus* induced early flowering, with increases in seed and flower sizes, but reductions in pollen fertility and seed setting rates. Additionally, *ljagd2a* mutation resulted in increased expression of nodulin genes and corresponding increases in infection threads and nodule numbers following inoculation with *R**hizobium*. Changes in expression of *LjAGD2a* in *L. japonicus* also affected endogenous SA contents and hence resistance to pathogens. Our results indicate that LjAGD2a functions as an LL-DAP aminotransferase and plays important roles in plant development. Moreover, *LjAGD2a* activates defense signaling via the Lys synthesis pathway, thereby participating in legume–microbe interaction.

## 1. Introduction

The functional significance of lysine (Lys) catabolism in plants is being gradually elucidated. Three routes of Lys catabolism have been described in plants: the cadaverine, saccharopine and *N*-hydroxypipecolic acid (NHP) pathways [1,2]. In the cadaverine pathway, Lys decarboxylase catalyzes conversion of Lys into the alkaloid cadaverine, a precursor of a large family of plant alkaloids [3]. In the saccharopine pathway, Lys is converted into α-aminoadipate by lysine-ketoglutarate reductase/saccharopine dehydrogenase (LKR/SDH) and α-aminoadipate semialdehyde dehydrogenase. In this pathway (also called the α-amino adipic acid pathway), lysine is catabolized into acetyl-CoA and glutamate [4]. In the NHP pathway, Lys is converted into NHP by three reaction steps. The first, catalyzed by ALD1 (AGD2-like defense response protein 1) [1,2], deaminates Lys into dehydropipecolate (DHP). Then, systemic acquired resistance-deficient 4 (SARD4) reduces DHP to pipecolate, and finally flavin-dependent monooxygenase (FMO1) catalyzes *N*-hydroxylation of pipecolate, forming NHP [1]. Pipecolic acid (Pip) and NHP play key roles in plant immunity due to their role in the activation of systemic acquired resistance (SAR) upon pathogen attack [1,5,6].

The major pathway of Lys biosynthesis in prokaryotes is the succinyl and acetyl-DAP pathway [7], in which Asp semialdehyde is first converted into tetrahydrodipicolinate (THDPA) through the sequential action of dihydrodipicolinate synthase (DapA) and dihydrodipicolinate reductase (DapB). Then, THDPA is succinylated by a succinylCoA-dependent transferase (DapD), which results in opening of the ring and exposure of a keto group that serves as the acceptor site for the next reaction, a Glu-dependent transamination. At least two gene products, DapC and ArgD, have been shown to catalyze the transamination. After aminotransfer, the succinyl group is removed by a desuccinylase (DapE), forming LL-diaminopimelate (LL-DAP). An epimerase (DapF) then converts LL-DAP to m-DAP. Finally, formation of Lys from m-DAP is catalyzed by meso-diaminopimelate (m-DAP) decarboxylase (LysA) [8]. In plants such as *Arabidopsis thaliana*, an *LL*-DAP aminotransferase (LL-DAP-AT) can catalyze direct conversion of THDPA to *LL*-DAP by transamination, and the catalytic reaction can also be reversed, as shown using *O*-aminobenzaldehyde (OAB), which forms a colored dihydroquinazolinium adduct with THDPA [8,9]. *LL*-DAP-AT has also been called AGD2 (Aberrant Growth and Death2). AGD2 plays vital roles in plant development and defenses in Arabidopsis: T-DNA insertion mutation of AGD2 causes embryo lethality, and a point mutation of AGD2 causes changes including pointed leaves, reduced stature and elevated resistance to *Pseudomonas syringae* in Arabidopsis [8,10,11,12]. 

We previously reported that *LjALD1*, an orthologous gene of Arabidopsis *ALD1*, is highly expressed in root nodules and involved in nodulation through regulation of endogenous SA contents in *Lotus japonicus* [13]. In this study, we investigated the biochemical characteristics and biological functions of AGD2 related to Lys synthesis in *L. japonicus*. Our results show that LjAGD2a has *LL*-DAP-AT activity and plays diverse roles in *L. japonicus* development. Moreover, *LjAGD2a* positive regulates resistance to the bacterial pathogen *Ralstonia solanacearum* and negatively modulates nodulation in *L. japonicus* by maintaining the balance of Lys metabolism.

## 2. Results

### 2.1. Expression Pattern of LjAGD2a in L. japonicus

Both AGD2 and ALD1 are encoded by single genes in Arabidopsis, and their amino acid sequences have high homology. However, there are four LjAGD2 homologs in the *L. japonicus* genome. LjALD1 and LjALD1L belong to the same cluster, while LjAGD2a and LjAGD2b are members of another cluster (Appendix A). 

To investigate the function of *AGD2* in *L. japonicus*, expression patterns of *LjAGD2* genes were analyzed. Results of real-time PCR analysis showed that L*jAGD2a* and *LjAGD2b* were expressed in all the tested tissues (Appendix A). β-glucuronidase (GUS) staining in *pLjAGD2a:GUS* and *pLjAGD2b:GUS* plants corroborated the expression of *LjAGD2a* and *LjAGD2b* in the leaves, stems, roots, flowers, pods and nodules (Figure 1). In transverse sections, GUS signals were detected mainly in vascular tissues of root and leaf cells in both *pLjAGD2**b**:GUS* and *pLjAGD2b:GUS* plants (Figure 1D,F,L,N), but GUS expression was also present in the epidermis cells in *pLjAGD2**b**:GUS* roots (Figure 1L). Examination of nodule sections showed that both *LjAGD2a* and *LjAGD2a* were mostly expressed in vascular bundles (Figure 1F–H,M). These results suggest that *LjAGD2a* and *LjAGD2b* are largely expressed in the vascular tissues of the various plant organs in *L. japonicus*.

### 2.2. LjAGD2s Are Localized in Chloroplasts and Function as LL-Diaminopimelate Aminotransferases

To investigate where LjAGD2a might function in cells, LjAGD2a-eYFP (enhanced yellow fluorescent protein) and LjAGD2b-eYFP constructs were transiently expressed in Arabidopsis protoplasts. The results show that LjAGD2a and LjAGD2b were both located in chloroplasts (Figure 2), in accordance with their putative role in Lys synthesis as Lys is known to be synthesized in chloroplasts [14,15].

To determine the biochemical function of LjAGD2, CDSs of the LjAGD2a and LjAGD2b genes were inserted into an *Escherichia coli* expression vector. Molecular masses of tag-purified/cleaved recombinant proteins estimated by SDS-PAGE analysis matched the predicted sizes (in kDa) of LjAGD2a and LjAGD2b well (Figure 3A). Aminotransferase activities were assayed with these recombinant proteins using L-Lys and 2-oxoglutarate as substrates, with soluble protein extracts of *E. coli* containing vector alone as negative controls, and AtAGD2 as a positive control. The results showed that LjAGD2 proteins can catalyze formation of Glu from Lys and 2-oxoglutarate (Figure 3B and Appendix A), indicating that they are aminotransferases. To corroborate the conclusion that LjAGD2s are *LL*-DAP aminotransferase (*LL*-DAP-ATs) in plants, we used *dapD* and *dapE* mutants, which are auxotrophic for DAP in functional complementation assays [8]. The *dapD* and *dapE* mutant strains were transformed with either an empty plasmid or expression plasmids with LjAGD2a and LjAGD2b. The results showed that *dapD* and *dapE* mutant stains expressing *AtAGD2*, *LjAGD2a* or *LjAGD2b* can grow on medium without DAP, but not the negative controls (Figure 3C). In this experiment, *AtAGD2* was used as a positive control. Therefore, LjAGD2a and LjAGD2b can bypass the succinylation and desuccinylation reactions required by *E. coli* to synthesize *LL*-DAP from THDPA (Figure 3D), strongly indicating that LjAGD2a and LjAGD2b are aminotransferases involved in Lys biosynthesis. 

### 2.3. LjAGD2a Affects Plant Morphology and Development

To assess the effects of changing expression of *LjAGD2* on *L. japonicus*, we obtained an insertional mutant line *ljagd2a* (LORE ID: 30001979, Appendix A) and three *LjAGD2a* overexpression lines (designated *OeLjAGD2a-1*, *OeLjAGD2a-3* and *OeLjAGD2a-6*). The expression of *LjAGD2a* in each line was detected by semi-quantitative or quantitative PCR (Appendix A). We assayed *LL*-DAP-AT activity in leaves of the mutant and transgenic plants. The results showed that overexpression of *LjAGD2a* significantly increased, and the mutation of *LjAGD2a* significantly decreased *LL*-DAP-AT activity in *L. japonicus*. Overexpression of *LjAGD2a* in *ljagd2a* increased the transcription level of *LjAGD2a* and partially restored *LL*-DAP-AT activity (Figure 4A). 

Compared to wildtype controls, *ljagd2a* mutants showed retarded growth, delayed flowering and smaller seeds, while overexpression of *LjAGD2a* in *L. japonicus* resulted in dwarf shoots, early flowering and larger seeds (Figure 4B,C,F–H). Additionally, *OeLjAGD2a* lines had shorter pods, larger floral organs and thicker leaves (Figure 4C–F). Owing to the roles of *AtAGD2* in embryo development in Arabidopsis, we also assessed the effects of *LjAGD2a* on the seed setting rate and pollen fertility. No significant differences in seed setting rate and pollen fertility between the wildtype (Gifu B-129) and the *ljagd2a* mutant were observed (Figure 5). However, *OeLjAGD2a-3* and *OeLjAGD2a-6* plants had 60–75% lower seed setting rates than wildtype plants (Figure 5A,B). In addition, iodine-potassium iodide staining showed that wildtype pollen grains were round, whereas many *OeLjAGD2a* pollen grains were shrunken, irregular and wizened (Figure 5C). Nearly 100% of wildtype pollen grains became dark blue upon iodine staining, but the proportion of pollen grains from *OeLjAGD2a-3* and *OeLjAGD2a-6* was 35–50% (Figure 5D). After incubation on medium for 30 min at 37 °C, the germination rate of pollen from *OeLjAGD2a-3* and *OeLjAGD2a-6* plants was around 18–40%, whereas the rate of germination of wild type pollen was about 85% (Figure 5E,F). Clearly, overexpression of *LjAGD2a* substantially reduces the fertility of *L. japonicus* pollen. 

### 2.4. Alteration of LjAGD2a Expression in L. japonicus Changed Resistance to the Bacterial Pathogen R. Solanacearum 

Here, to investigate the function of *LjAGD2**a* in *L. japonicus*–pathogen interactions, we analyzed resistance phenotypes. Two-day-old seedlings were infected with *R. solanacearum* and its growth in the roots was monitored by counting the bacteria. We found no significant difference in densities of the bacteria in the overexpressing or mutant lines 1- and 3-days post-infection (dpi), but growth of *R. solanacearum* was dramatically inhibited on *OeLjAGD2a* roots at 5 dpi (Figure 6A). *Ljagd2a* was consistently more susceptible to *R. solanacearum* than its wildtype (Figure 6A,B). We also assayed the growth of *R. solanacearum* on detached leaves, and the results showed that *OeLjAGD2a* plants were more resistant and *ljagd2a* plants more susceptible to *R. solanacearum* than wildtype plants (Figure 6C). Simultaneous measurements of SA levels in leaves after infection with *R. solanacearum* showed that overexpression of *LjAGD2a* caused increases in SA contents in the leaves while the mutation of *LjAGD2a* resulted in decreased SA contents (Figure 6D,E). In view of divergent roles of Arabidopsis *AGD2* homologs in defense signaling [14], we further investigated the resistance phenotype of *ljald1* mutants, and the results showed that *ljald1* mutants were more susceptible to *R*. *solanacearum* than wildtype plants (Appendix A). 

### 2.5. Knockout Mutation of LjAGD2a Increased Infection Thread and Nodule Numbers in L. japonicus

Next, nodulation phenotypes were observed. Compared to the wildtype, *ljagd2a* mutants had more infection threads and nodules after inoculation with *Mesorhizobium loti* (Figure 7A,B). In addition, nodules were smaller in *ljagd2a* plants than in the wildtype plants (Figure 7D). Although *OeLjAGD2a* plants had few infection threads at 3 and 7 days after *M. loti* inoculation, the number of infection threads at 14 days and nodule numbers at 2- and 4-weeks post-infection did not significantly differ from numbers in wildtype plants (Figure 7A–C). Therefore, overexpression of *LjAGD2a* may only delay infection of *M. loti*. Subsequent assessment of the expression of nodulin genes after *M. loti* infection showed that levels of *LjENOD40*, *LjNIN* and *LjERN1* transcripts were significantly higher in *ljagd2* plants but not significantly different in *OeLjAGD2a* lines, compared to their wildtype (Figure 8). 

## 3. Discussion

Our experiments with *ljagd2a* insertion mutants and *OeLjAGD2a* plants showed that the Lys biosynthesis pathway plays an important role in the development and microbial interactions of *L. japonicus* (and likely other leguminous plants). The insights obtained are summarized in this section.

### 3.1. LjAGD2s Function as LL-DAP-Ats

AGD2 in Arabidopsis has *LL*-DAP aminotransferase (LL-DAP-AT) activity, which can directly convert THDPA to *LL*-DAP and is equivalent to the functions of DapD, DapC and DapE in prokaryotes [8]. Recombinant LjAGD2a and LjAGD2b have aminotransferase activity in vitro (Appendix A) and functional complementation of LjAGD2a and LjAGD2b in *dapD* and *dapE* mutants showed that LjAGD2s can function in the ‘forward direction’, towards synthesis of *LL*-DAP in *E. coli.* These results suggest that LjAGD2s have *LL*-DAP-AT activity. Alteration of *LjAGD2a* expression in *L. japonicus* also changed the *LL*-DAP-AT activity, corroborating LjAGD2s’ functionality as aminotransferases involved in the Lys biosynthetic direction in *L. japonicus*. In addition, LjAGD2s, such as AtAGD2, are located in the chloroplast, which is consistent with their roles in Lys synthesis in chloroplasts [14,16]. Therefore, we propose that LjAGD2s have the same biochemical functions in plants as AtAGD2.

### 3.2. LjAGD2s May Have Functional Redundancy in Regulation of Plant Development

Lys is an important signaling amino acid that regulates plant growth and responses to the environment [17]. In Arabidopsis, loss of *AGD2* function renders embryos inviable [14], and a point mutation of *agd2* leads to metabolic disorders and growth inhibition [12]. Here, we found that there are two *AGD2* genes in *L. japonicus* and knockout of *LjAGD2a* both reduced *LL*-DAP-AT activity and suppressed growth of shoots. However, *ljagd2a* plants did not have clear abnormalities in embryo development, seeds or fertility (Figure 5). Therefore, we hypothesize that LjAGD2a and LjAGD2b may be functionally redundant in regulation of plant development. Reductions in *LL*-DAP-AT activity cause changes in levels of amino acids and other metabolites involved in Lys synthesis and related pathways [12,18,19], thus resulting in metabolic disorders that affect development of *L. japonicus*.

### 3.3. AGD2a Affects Disease Resistance and Nodulation via the Lys Biosynthesis Pathway

*AGD2* suppresses SA-dependent defenses while *ALD1* increases SA contents and defense signaling in Arabidopsis [10,14]. We found that *ljald1* mutants were more susceptible than wildtype plants to *R. solanacearum* (Appendix A). However, contrary to the negative effect of *AtAGD2* on defenses, knockout mutation of *LjAGD2a* reduced resistance of *L. japonicus* to the bacterial pathogen while its overexpression significantly increased resistance (Figure 6). We speculate that *LjAGD2* is involved in plant disease resistance through its participation in Lys biosynthesis and hence regulation of Lys supply and associated metabolites. Similarly, we found elevated numbers of infection threads and nodules in *ljagd2* plants following infections with *M. loti*, as in *LjALD1* RNAi plants (Figure 7) [13]. Overexpression of salicylate hydroxylase *NahG* in *M. truncatula* and *L. japonicus* resulted in the reduction of endogenous SA levels and therefore a greater number of infection threads and nodules [19]. Consistent with this result, knockdown of *LjALD1* decreased SA content to promote nodulation in *L. japonicus* [13]. We thus speculate that *ljagd2*, like *ljald1*, can affect Lys metabolism by attenuating SA-dependent defense signaling, and hence expression of nodulin genes. Small nodules in *ljagd2a* plants may result from an insufficient carbon source supply from host plants since mutation of *AGD2* in Arabidopsis can reportedly cause a shift in amino acid metabolism and C/N imbalance [12,20].

## 4. Materials and Methods

### 4.1. Plant Growth Conditions and Treatments

Two ecotypes of *L. japonicus*, Gifu B-129 and MG-20, were mainly used in this study. LORE1 insertion mutant lines of *ljagd2a* (plant IDs in Lotus Base: 30001979) with an *L. japonicus* Gifu B-129 background were obtained from Lotus Base [21,22] (https://lotus.au.dk/, accessed on 11 November 2020). *pLjAGD2a:GUS* and *pLjAGD2b:GUS* lines were obtained by transformation of MG-20 with *Agrobacterium tumefaciens* strain AGL1. Plants were grown in growth chambers with the temperature maintained at 22 °C (light intensities about 100 µm m^−2^ s^−1^). To germinate seeds, they were treated with concentrated sulfuric acid for 7 min, sterilized with 1.5% sodium hypochlorite solution for 10 min, washed six times with sterile water and placed in sterile dishes for 2 d. To induce production of infection threads, 2-day-old seedlings were moved to 0.8% agar plates containing 1/2 B&D solution (Broughton and Dilworth 1971) for 2 days, then infected by adding 1 mL suspension of *DsRed*-labeled *Mesorhizobium loti* (OD_600_ = 0.01) and the excess liquid was sucked off 5 min later. The materials were then cultured in an artificially illuminated growth cabinet providing 22 °C, 16 h light and 18 °C, 8 h dark cycles with a constant relative humidity of 70% (GXZ-380, Ningbo Jiangnan Instrument Factory). Infection threads on the whole root were counted at different time points after inoculation with *M. loti* expressing *DsRED* under a fluorescence microscope (DMI4000B, Leica, Germany). Rhizobial strain was *Mesorhizobium loti* MAFF303099 and cultured in YM liquid medium (containing 10 g/L Mannitol, 3 g/L Yeast extract, 0.25 g/L K_2_HPO_4_, 0.2 g/L MgSO_4_7H_2_O, 0.1 g/L NaCl, pH 6.8). YM liquid medium was added to 15 g/L agar to produce the solid medium.

### 4.2. Plasmid Constructs and Plant Transformation

Full-length *LjAGD2a* and *LjAGD2b* was amplified by PCR with the primers shown in Appendix A using *L. japonicus* cDNA as a template. *LjAGD2a* fragments were digested with *Nco* I and *Eco*R I, then cloned into *pSAT6-**eYFP-N1* behind the CaMV 35S promoter to form *35S::LjAGD2a-**eYFP*.

Full-length *LjAGD2a* and *LjAGD2b* or fragments lacking signal peptides were amplified by PCR, again with primers shown in Appendix A and using *L. japonicus* cDNA as template, digested with *Nco* I and *Xho* I, and cloned into *pGEX-KG*. The resulting recombinant plastids for functional complementation were transformed into *E. coli* strains *dap AT980* (*dapD*) and *dap AT984* (*dapE*) (The Coli Genetic Stock Center).

For construction of the *pLjAGD2a:GUS* and *pLjAGD2b:GUS* vector, a sequence upstream of the first initial ATG codon of *LjAGD2a* and *LjAGD2b* was amplified, respectively. The specific primer sequence is shown in Appendix A. After digestion by restriction enzymes, the cDNA fragments were cloned into *pCAMBIA1391Z* before the *GUS* gene. The resulting construct was introduced into *Agrobacterium tumefaciens* strain AGL1 by the freeze–thaw procedure. *L. japonicus* was transformed according to a previously described method [13]. Hypocotyls excised from *L. japonicus* MG-20 seedlings were infected with *A. tumefaciens* strain AGL1 harboring the *pLjAGD2a:GUS* or *pLjAGD2b:GUS* constructs. Regenerated plants showing hygromycin resistance were grown in vermiculite pots to harvest T1 seeds. 

### 4.3. Functional Complementation of E. coli Dap Mutants and Enzyme Assays

*Dap* mutants were cultured in LB liquid medium containing DAP and 100 μg/mL ampicillin until their absorbance reached 0.4 to 0.6. Appropriate volumes of bacterial suspension were diluted 100-fold, then 10 μL of diluted bacterial solution was spread on solidified NZY medium (5 g/L NaCl, 2 g/L MgSO_4_·7H_2_O, 10 g/L casein hydrolysate, 5 g/L yeast extract, 15 g/L agar) supplemented with 0.2% (*w/v*) Ara and 100 μg/mL ampicillin with or without 50 mg/mL DAP. All plates were incubated at 37 °C for 12 h. 

Recombinant plastids for functional complementation were also used for enzyme assays in vitro. For protein expression and purification, recombinant plastids were transformed into *E. coli* Rosetta strains. The resulting strains were grown on LB medium at 37 °C to OD_600_ = 0.5, then protein expression was induced by incubation with 0.1 mM isopropylthio-β-galactoside for 12 h at 20 °C. Cells were lysed by sonication in PBS buffer (8 g/L NaCl, 0.2 g/L KCl, 1.44 g/L Na_2_HPO_4_, 0.24 g/L KH_2_PO_4_, pH 8.0) with 1 mM PMSF. After centrifugation at 4 °C and 12,000 *g* for 30 min, the supernatant was filtered with a 0.45 μM filter, then added to 1 mL GST resin. The soluble fraction was washed three times with PBS buffer. Freshly prepared 10 mM reduced glutathione solution in 50 mM Tris-HCl, pH 8.0, was used to elute the target protein. The reaction system consisted of 50 mM Tris HCl (pH 8.0), 50 mM Lys, 50 mM 2-oxoglutarate (2-OG), 1 μM pyridoxal-5-phosphate, 100 mM MgCl_2_ and purified protein. After reaction at 37 °C for 6 h, anhydrous ethanol was added to a final concentration of 80% (v/v). After standing for 30 min, samples were centrifuged at 12,000 *g* for 2 min. The supernatant was dried, dissolved in 400 μL 0.01 M HCl and filtered through a 0.45 μm filter. Then, Glu was measured in 50 μL portions of the filtrate using a S-430D amino acid analyzer (Sykam, Germany), mobile phase of lithium citrate with pH sequentially rising from 2.9 to 4.2 and 8.0 (flow rate: 0.45 mL/min) and derivatization reagent flow rate of 0.25 mL/min. The temperatures of the column, post-column reaction equipment and automatic sampler were set at 38, 130 and 5 °C, respectively, and the eluate was monitored at 570 and 440 nm [23].

To assay enzyme activities in crude protein extracts, leaves were ground in liquid nitrogen, soluble proteins were dissolved in 100 mM HEPES.KOH (pH 7.6), and the resulting mixture was centrifuged at 12,000 *g* for 15 min, then subjected to buffer exchange using an Amicon Ultra 30,000 MWCO filter. The standard curve of protein concentration was obtained by the BSA method when A280 was used, and the extracted leaf protein concentration was obtained by standard curve. A 200 μL reaction system containing 100 mM HEPESKOH (pH 7.6), 0.5 mM amino donor *LL*-DAP, 2.0 mM 2-OG, 0.25 mg *O*-aminobenzaldehyde (OAB), and crude soluble protein was used. Reactions were incubated at 30 °C and the ΔA (440 nm) measured [24].

### 4.4. Phylogenetic Tree Construction

Multiple sequence alignments of amino acid sequences of LjAGD2s and other AGD2 and ALD1 orthologs in different plant species were performed using Clustalx 2.1 and DNAMAN 6.0 software. A phylogenetic tree was constructed by MEGA 6.0 using the neighbor-joining (NJ) method and 1000 bootstraps [25,26].

### 4.5. RNA Isolation and Expression Analysis

Total RNA was extracted from selected *L. japonicus* tissues using a HiPure Plant Mini Kit (Guangzhou Magen Biotechnology Co., Ltd., Guangzhou, China) following the manufacturer’s instructions, and the isolated RNA was treated with RNase-free Dnase I (Magen). RNA samples (2 μg) were used for reverse transcription with a GoScript™ Reverse Transcription System (Promega, Madison, WI, USA). Quantitative real-time PCR (qRT-PCR) analysis was performed using the primers listed in Appendix A, with *LjATPase* and *LjUBC* genes as internal controls. qRT-PCR was performed with a LightCycler 480 (Roche Scientific, Basel, Switzerland) and the following amplification conditions: 10 min at 95 °C, followed by 40 cycles of 95 °C for 5 s, 60 °C for 20 s and 72 °C for 20 s. The experiment was performed with three replicates and average measurements are presented.

### 4.6. Subcellular Localization of LjAGD2s

Protoplasts of Arabidopsis ecotype Columbia were isolated and transformed following a previously described method [26]. Fluorescence images were recorded with a META LSM510 laser scanning confocal microscope (Zeiss^10^) using a 63 × 1.40 oil objective. eYFP fluorescence was imaged at an excitation wavelength of 514 nm (30% power) and 527 nm emission wavelength. The autofluorescence of chloroplasts was observed using an excitation wavelength of 543 nm and emission wavelength of 562 nm.

### 4.7. Histochemical Staining and Microscopic Analysis

For histochemical assays, seedlings of *pLjAGD2a::GUS* or *pLjAGD2b::GUS* transgenic lines at selected growth stages were stained with GUS staining buffer (1 mM X-Gluc, 0.5 mM K_3_[Fe(CN)_6_], 0.5 mM K_4_[Fe(CN)_6_], 1 mM EDTA, pH 8.0, 0.1 mM PBS buffer, pH 7.0) at 37 °C for 6–8 h after vacuum infiltration for 15 min. For nodule sectioning, 2-week-old nodules were harvested after inoculation with *Ds*Red-labeled *M. loti*, and sections (50 μm) of fresh nodules were cut with a Leica VT1200 S vibrating-blade microtome.

For the analysis of leaf thickness, leaves were fixed with 2% agar, and 50 μm slices were obtained, in 0.1× PBS buffer (8 g/L NaCl, 0.2 g/L KCl, 1.44 g/L Na2HPO_4_, 0.24 g/L KH_2_PO_4_, pH 8.0), using a Leica VT1200 S vibrating-blade microtome.

### 4.8. Observation of Pollen Morphology and Germination

For observations of pollen morphology and fertility, pollen grains were harvested from fully expanded flowers. After staining with an iodine–potassium iodide solution, they were viewed with a bright-field microscope. To assess the pollen germination rate, pollen grains were dispersed in a solution of 10% sucrose, 0.01% H_3_BO_3_ and 0.05% Ca(NO_3_)_2_·4H_2_O. After incubation at 37 °C for 30 min, germinated pollen grains were viewed with a bright-field microscope [27]. 

### 4.9. Cultivation of Bacteria and Plant Inoculation

*Ralstonia solanacearum* strain FQY_4 was cultured in SMSA liquid medium (containing 2.5 mL 1% polymyxin, 125 μL 1% crystal violet, 1.25 mL 1% TTC, 625 μL 1% bacitracin, 125 μL 0.1% penicillin, 125 μL 1% chloramphenicol and 5 μg/mL gentamicin per liter) to OD_600_ = 0.5–0.6, then 100 μL bacterial liquid was coated on SMSA solid medium and grown for 2 days [28]. The bacteria were suspended in sterile 10 mM MgCl_2_ to OD_600_ = 0.001. Portions (100 μL) of *R. solanacearum* suspension were injected into the back of fresh leaves (*n* > 15) of selected plant lines by a 1 mL sterile syringe. After inoculation, the leaves were stored in agar medium containing 10 ppm imidazole. After cultivation in the dark at 28 °C for 2 days, the leaves were exposed to light for 16 h at 22 °C and dark for 8 h at 18 °C for 2 days. Leaf discs were made with a puncher, washed with sterile water five times and ground with a pestle in 10 mM MgCl_2_. The resulting homogenates were fully mixed and continuously diluted. A 100 μL diluted bacterial solution was spread on SMSA solid medium, and the number of *R. solanacearum* colonies was counted 48 h later.

### 4.10. Measurement of Salicylic Acid and Free Amino Acid Contents

Salicylic acid was extracted as previously described [13]. For the extraction of amino acids, 1 g samples of leaves were taken just before the beginning and end of the photoperiod from the same position of each plant line, ground into homogenate with 10 mL 0.1 N HCl and agitated ultrasonically at 4 °C for 15 min. After a further hour of incubation, the homogenate was centrifuged at 3000× *g* for 30 min, and then absolute ethanol was added to the supernatant to a final concentration of 80% to precipitate protein. After further centrifugation at 3000× *g* for 30 min, the supernatant was transferred to another tube and dried under vacuum. The dry matter was dissolved in 400 μL of 0.01 N HCl, and the resulting solution was filtered using a 0.45 μm filter membrane. Finally, 50 μL portions of the filtrate were subjected to analysis with an automatic amino acid analyzer (Sykam S-430D, Munich, Germany) using the C4 free 41AA standard procedure, an LCA K07/Li 4.6 × 150 mm chromatographic column and analyte detection wavelengths of 570 and 440 nm.

### 4.11. Statistical Significance

The numbers of independent repetitions of each experiment are given in the corresponding figures and tables. Statistical significance was tested as described in the given figures and tables. For all assays, differences between samples or time points with probabilities < 0.05 and 0.01 are indicated in figures by single and double asterisks, respectively. SPSS Statistics V17.0 software was used for all statistical analyses.

## 5. Conclusions

Our results indicate that LjAGD2a functions as an LL-DAP aminotransferase. We examined how mutation of *LjAGD2a* and overexpression of *LjAGD2a* affected shoot, leaf, flower, pod, seed and pollen development, which suggests that *LjAGD2a* plays important roles in plant development. *LjAGD2a*, like *LjALD1*, has a positive role in plant–pathogen interaction and a negative role in legume–rhizobia symbiosis. Our results also indicate that *LjAGD2a* activates defense signaling via the Lys synthesis pathway, thereby participating in legume–microbe interaction.

## Figures and Tables

**Figure 1 ijms-23-06863-f001:**
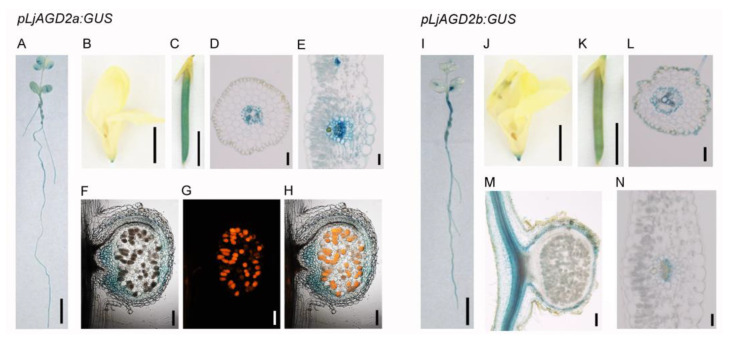
Expression pattern of *LjAGD2s* in *L. japonicus*. (**A**–**H**) GUS histochemical staining of *pLjAGD2a:GUS* plants. (**A**) GUS histochemical staining of a 2-week-old *pLjAGD2a:GUS* seedling. (**B**) GUS staining of a flower of a *pLjAGD2a:GUS* plant. (**C**) GUS staining of a pod of a *pLjAGD2a:GUS* plant. (**D**) Section of a root of a 2-week-old *pLjAGD2a:GUS* plant. (**E**) Section of a leaf of a 2-month-old *pLjAGD2a:GUS* plant. (**F**–**H**) Section of a nodule of a *pLjAGD2a:GUS* plant 2 weeks after inoculation with *Mesorhizobium loti* expressing *DsRED*. GUS histochemical staining of the nodule in (**F**), Bacteriods in the infection cells expressing red fluorescence in (**G**), Overlap of F and G images in (**H**). (**I**–**N**) Histochemical staining of *pLjAGD2b:GUS* plants. (**I**) GUS histochemical staining of a 2-week-old *pLjAGD2b:GUS* seedling. (**J**) GUS staining of a flower of a *pLjAGD2b:GUS* plant. (**K**) GUS staining of a pod of a *pLjAGD2b:GUS* plant. (**L**) Section of a root of a 2-week-old *pLjAGD2b:GUS* plant. (**M**) Section of a nodule of a 2-week-old *pLjAGD2b:GUS* plant inoculated with *M. loti*. (**N**) Section of a leaf of a 2-month-old *pLjAGD2b:GUS* plant. Sixty seedlings were subjected to GUS staining as illustrated in (**A**,**I**), with more than 90% showing the phenotype. Flowers and pods collected from more than 10 individual plants were used for the GUS staining illustrated in (**B**,**C**,**J**,**K**). For sections, more than six samples were used.

**Figure 2 ijms-23-06863-f002:**
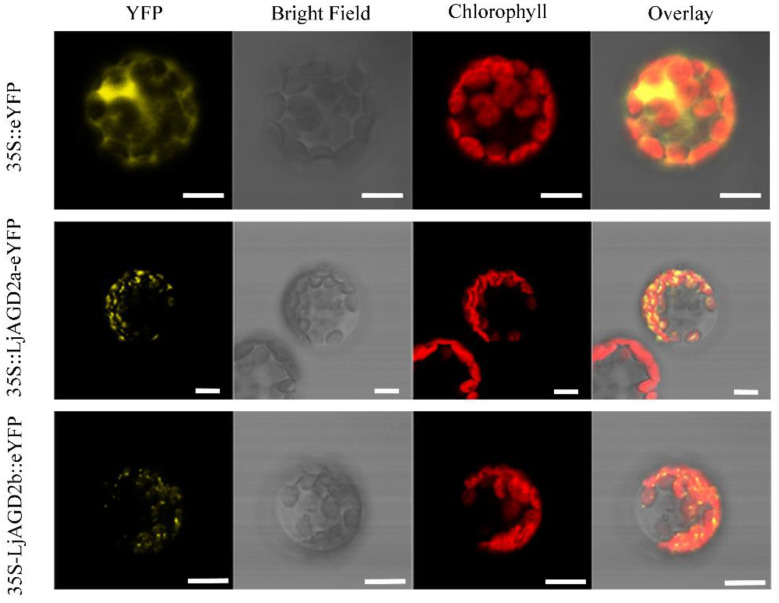
Subcellular localization of LjAGD2a and LjAGD2b. LjAGD2a-eYFP and LjAGD2b-eYFP images indicating the localization of LjAGD2a and LjAGD2b in the chloroplasts of Arabidopsis protoplasts. The top row of images indicates the control. Scale bar = 10 µm.

**Figure 3 ijms-23-06863-f003:**
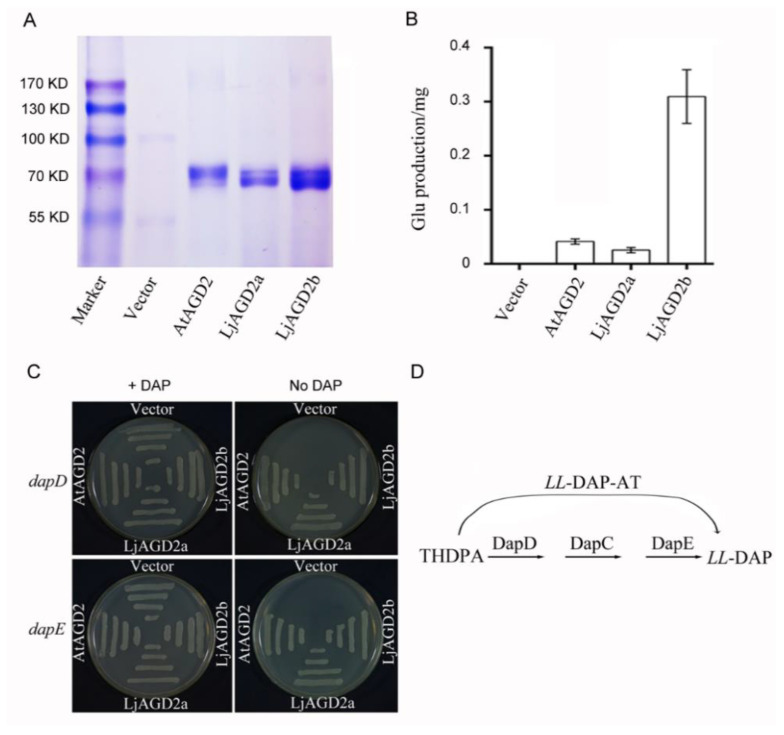
Aminotransferase activities of AGD2s in *E. coli*. (**A**) Electrophoretic patterns of recombinant proteins purified using a Ni^2+^-nitrilotriacetic acid column. (**B**) Aminotransferase activities of AGD2s, assayed using 50 mM Lys and 50 mM 2-oxoglutarate as substrate and co-substrate, respectively. Aminotransferase activity was determined by measuring the concentration of the reaction product Glu. Bars indicate standard deviations (*n* = 3). (**C**) Complementation of *dap* mutants with AGD2. Strains AT980 (*dapD* mutant) and AT984 (*dapE* mutant) were transformed with either the plasmid vector (pGEX-KG) or an LjAGD2a, LjAGD2b and AtAGD2 (At4g33680) expression plasmid (pGEX-KG-AGD2). Colonies were selected on LB medium with 50 µg/mL DAP and 100 µg/mL ampicillin. Individual colonies were then plated onto NZY medium supplemented with 0.2% (*w*/*v*) Ara with or without 50 µg/mL DAP. The cultures were grown at 37 °C for 12 h. (**D**) Diagram of the DAP pathway in *E. coli* with the reaction catalyzed by *LL*-DAP-AT indicated.

**Figure 4 ijms-23-06863-f004:**
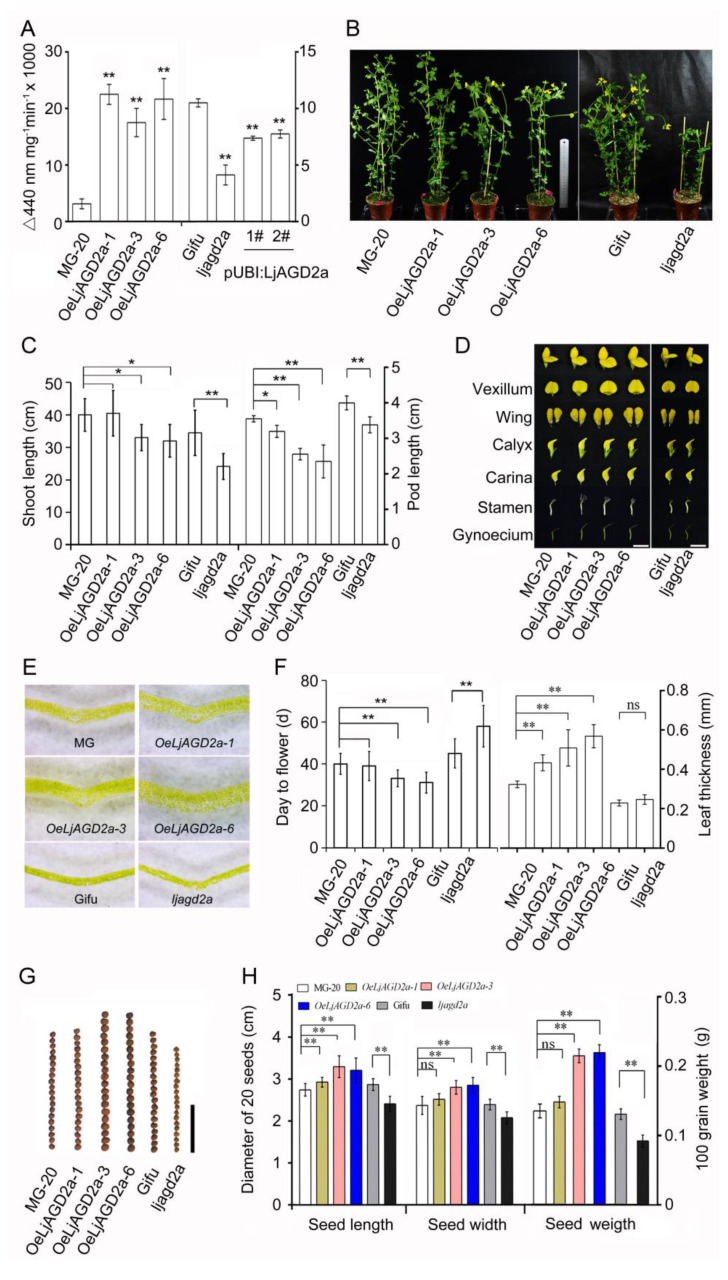
Morphological effects of changing expression of *LjAGD2a* in *L. japonicus*. (**A**) *LL*-diaminopimelate aminotransferase (*LL*-DAP-AT) activities, showing a drastic reduction in *ljagd2a* and a significant increase in *OeLjAGD2a* lines (means with error bars indicating SD of five biological replicates). (**B**) Representative photograph of 8-week-old plants showing that *OeLjAGD2a* and *ljagd2a* plants had shorter roots than wild type plants (scale bar = 10 cm). (**C**) Shoot lengths, showing retarded growth of *OeLjAGD2a* and *ljagd2a* lines (**Left**). Both *OeLjAGD2a* and *ljagd2a* lines also produced shorter pods than wildtype plants (**Right**). (**D**) Photograph of 10-week-old plants showing that *OeLjAGD2a* plants had bigger flowers than wildtype plants, but insertional mutation of *LjAGD2a* did not change the flower size. Scale bar = 1 cm. (**E**) Sections showing that *OeLjAGD2a* plants had thick leaves. Scale bar = 1 mm. (**F**) *OeLjAGD2a* lines exhibited early flowering while insertional mutation of *LjAGD2a* delayed flowering (**Left**). Bar chart confirming that leaves of *OeLjAGD2a* lines were significantly thicker than wildtype leaves, but *ljagd2a* leaves did not significantly differ from wildtype leaves in thickness (**Right**). (**G**) Effects of changing *LjAGD2a* expression on seed morphology. Scale bar = 1 cm. (**H**) Seed sizes showing that *LjAGD2a* overexpression led to larger seeds, while *ljagd2a* seeds were smaller than wildtype seeds (**Left**). Measurement of 100-kernel weights of seeds showing that *LjAGD2a* overexpression and mutation resulted in increases and reductions in seed weight, respectively (**Right**). In the tests used to obtain results presented in (**C**,**F**,**H**), *n* = 20–30 per treatment. *, ** and ns indicate *p* < 0.05; *p* < 0.01 and no significant difference, respectively, according to the Duncan test. Error bars indicate the SD of three biological replicates.

**Figure 5 ijms-23-06863-f005:**
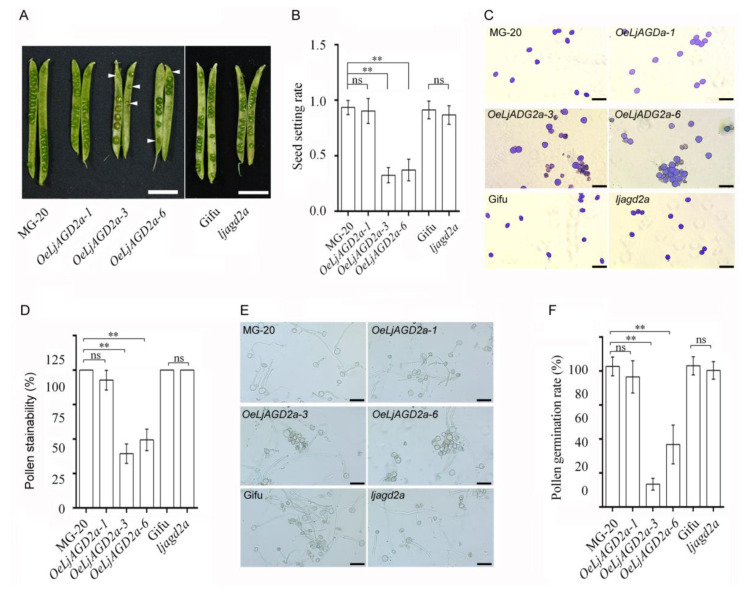
Effects of changing expression of *LjAGD2a* on fertility in *L. japonicus*. (**A**) Immature seeds in *OeLjAGD2a* siliques. Aborted dead seeds are indicated by arrows. Scale bar = 1 cm. (**B**) Measured seed setting rates, which were dramatically decreased in *OeLjAGD2a* plants, relative to wildtype rates, but not significantly altered in *ljagd2a* mutants (means obtained from analyses of more than 50 siliques with standard deviations: ** indicate *p* < 0.01, respectively, according to the Duncan test). (**C**) Photograph showing morphology of pollen grains. (**D**) Stainability of pollen by iodine-potassium iodide solution, indicating viability. (**E**) Photograph showing results of in vitro pollen germination assays. (**F**) Pollen germination rates, based on analyses of pollen grains from 5 to 7 anthers of each of 8 to 10 plants (** and ns indicate *p* < 0.01 and no significant difference, respectively, according to the Duncan test). Scale bar = 50 µm in (**C**,**D**). Error bars in (**B**,**D**,**F**) indicate the SD of three biological replicates.

**Figure 6 ijms-23-06863-f006:**
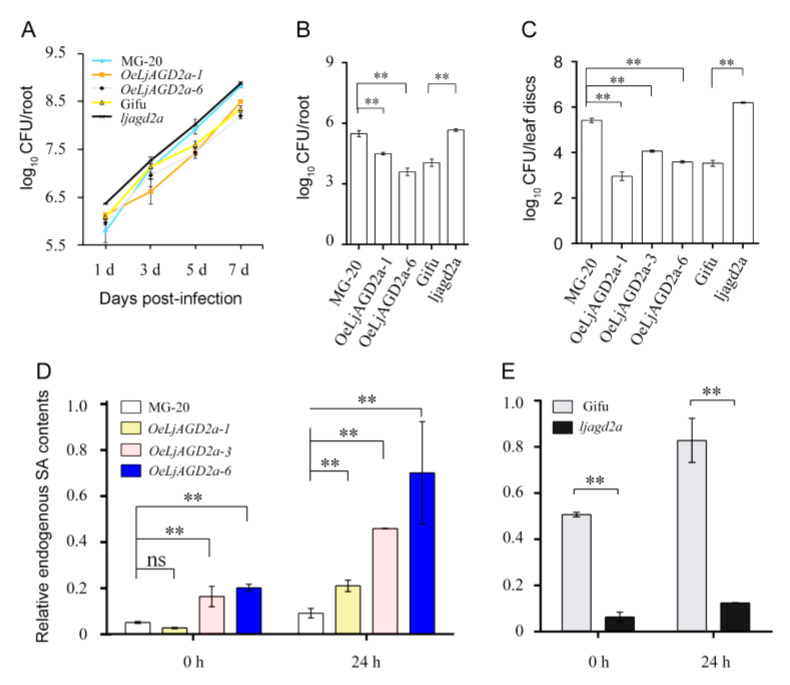
*LjAGD2a* promotes disease resistance in *L japonicus*. (**A**) Growth curves of *R. solanacearum* strains on the roots of transgenic *ljagd2a* lines, mutant lines and their respective wildtypes (MG-20 and Gifu129). Plants were infected with *R. solanacearum* at indicated time points. *R. solanacearum* grew more rapidly in *ljagd2a* plants than in their wildtype. In the first 3 days of pathogen infection, there was no significant difference in growth of *R. solanacearum* between *OeLjAGD2a* and MG-20 plants, but the pathogen’s growth was inhibited in *OeLjAGD2a* plants at 5 dpi. (**B**) Numbers of colony forming units (CFU) showing the disease susceptibility of roots. The indicated genotypes were inoculated with *R. solanacearum* by including the bacterium at a density equivalent to OD_600_ = 0.0002 in the growth medium and assaying the bacteria at 5 dpi. (**C**) Numbers of colony forming units (CFU) showing the disease susceptibility of detached leaves. Overexpression of *LjAGD2a* suppressed and its insertion mutation promoted growth of the pathogen. Detached leaves from 8-week-old plants were infected with *R. solanacearum*. (**D**) Total SA levels in *OeLjAGD2a* plants 0 and 24 h post-infection. (**E**) Total SA levels in *ljagd2a* mutants 0 and 24 h post-infection. Overexpression of *LjAGD2a* increased and its mutation reduced total SA levels after pathogen infection. 6–10 roots/leaves were used in each replicate and Error bars represent the SD of three replicates per genotype in (**A**–**C**). In experiments providing results shown in (**D**,**E**) each sample had three to four replicates and error bars represent the SD of 3–4 replicates. In A–E, ** indicates *p* < 0.01 according to the Duncan test.

**Figure 7 ijms-23-06863-f007:**
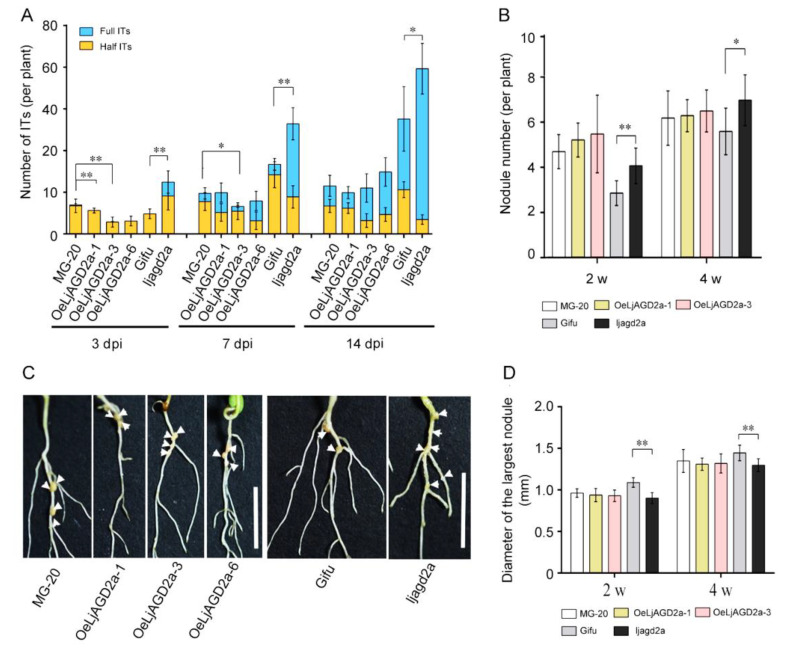
Changes in *LjAGD2a* expression affect *L japonicus* nodulation. (**A**) Numbers of infection threads (ITs) in transgenic plants and insertion mutants. Consistently higher numbers of ITs, relative to numbers in wildtype plants, were found in *ljagd2a* plants after *M. loti* inoculation. We also found fewer ITs in *OeLjAGD2a* plants at 3 days post-infection (dpi), but not 7 or 14 dpi. (**B**) Nodule numbers were higher in *ljagd2a* plants, but unchanged in *OeLjAGD2a* lines, relative to numbers in wildtype plants. (**C**) Nodules on roots 2 weeks after inoculation of plants. White arrows indicate nodules on the roots. (**D**) Measurements of the largest nodules on plants 4 weeks after inoculation with *M. loti* showing that mutation of *LjAGD2a* reduced nodule size. 15–20 plants were used in each replicate and three replicates per genotype were used in experiments providing data presented in A, B and D (±SD). * and ** indicate *p* < 0.5 and *p* < 0.01, respectively, according to the Duncan test.

**Figure 8 ijms-23-06863-f008:**
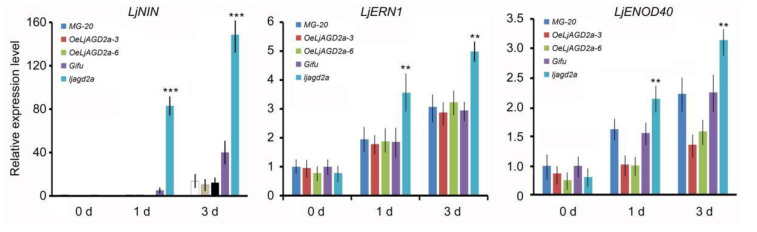
Levels of transcript of nodulin genes—the nodule inception gene (*LjNIN*), *ERF* required for nodulation 1 (*LjERN1*) and early nodulin gene (*LjENOD40*)—in *OeLjAGD2* plants and mutants. The expression of *LjNIN*, *LjERN1* and *LjENOD40* was increased in *ljagd2a* mutants after inoculation with *M. loti*, but overexpression of *LjAGD2a* did not change the expression of nodulin genes. The data were normalized using expression of *Actin* as an internal control. The Actin-normalized value for the 0-day wildtype sample was set to 1. Presented data are averages of triplicate RNA preparations (*n* = 15, ±SD). ** and *** indicate *p* < 0.01 and *p* < 0.001, respectively, according to the Duncan test.

## Data Availability

Not applicable.

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
