# Peer review of "Roles of AGD2a in Plant Development and Microbial Interactions of Lotus japonicus"

_ijms, 2022, doi:10.3390/ijms23126863_

Round 1
Reviewer 1 Report
Please see the attached .pdf file.

Author Response
Summary Huang et al. worked to identify the role of AGD2a (and to a lesser extent, AGD2b) in Lotus japonicus. They identified where these genes are expressed in the plant and where they are expressed within cells. They also demonstrated that these genes can complement dap mutations in E. coli, which indicates that they have aminotransferase activity. They then examined how mutation of LjAGD2a and overexpression of LjAGD2a affected shoot, leaf, flower, pod, seed, and pollen development. They also investigated the role of LjAGD2a in resistance to the plant pathogen Ralstonia solanacearum and its role in nodulation by Mesorhizobium loti.
Comments or Suggestions for Authors
I found 1 major issue and 3 moderate issues with your manuscript. The good news is that I don't see any need for additional experiments. The major issue is that you claim to see "wide" GUS staining in many tissues. I do not see that. The GUS staining is clearly limited to the vascular tissue. Fortunately, this does not affect any of your other conclusions, so all you need to do is fix your discussion of these experiments. The first moderate issue is that you do not identify the M. loti and R. solanacearum strains that you used. The second moderate issue is that there are several things you can do to improve the way that you present your data in the figures, including the descriptions in the figure captions. The third moderate issue is that you said you cloned the AGD2a ORFs, but in reality you cloned the CDSs. For more details, see below.
Major Issues 1. Misinterpretation of Figure 1 On p. 2, lines 84–93, you misinterpret the results of the GUS staining (Figure 1). Figures 1D and 1J do not show "wide" GUS expression in the root cells, they only show GUS expression in the stele (both) and in the epidermis (only in Figure 1J). Figures 1F and 1L do not show "wide" GUS expression in the leaf cells, they only show GUS expression in the vascular bundle. Figures 1E and 1K do notshow "wide" GUS expression in cortical cells, vascular bundles, and infection zones; they only show expression in the vascular bundles. Also, determinate nodules (like those of Lotus) do not have infection zones. The portion of the determinate nodule that contains bacteria is called the "nitrogen fixation zone" or sometimes the "infected zone", but this is distinct from the "infection zone" of indeterminate nodules. The nitrogen fixation zones in Figures 1E and 1K look dense, because of the abundance of symbiosome membranes, but they do not show any GUS expression. You need to revise your discussion of these images with the correct interpretation: LjAGD2a and LjAGD2b are only expressed in the vascular tissues of the various plant organs (with the exception of LjAGD2b, which is also expressed in the root epidermis).
Response: Thank you so much for your careful review! We revised the results in the revised manuscript from Line 89-96 according to your comments as “ In transverse sections GUS signals were detected mainly in vascular tissues of root and leaf cells in both pLjAGD2b:GUS and pLjAGD2b:GUS plants (Fig. 1D, F, L, N), but GUS expression was also present in the epidermis cells in pLjAGD2b:GUS roots (Fig. 1L). Examination of nodule sections showed that both LjAGD2a and LjAGD2a were mostly expressed in vascular bundles (Fig. 1F, 1G, 1H, 1M). These results suggest that LjAGD2a and LjAGD2b are largely expressed in the vascular tissues of the various plant organs in L. japonicus.”. And we also changed some images such as Fig1F, G, H, M.
Moderate Issues
- Bacterial Strain Identification In several places, including the methods (p. 11, lines 341–347 and p. 13, lines 430–432), you mention inoculating with Mesorhizobium loti, but you do not say which strain you used. If you used strain MAFF 303099 you will need to update the name to Mesorhizobium japonicum (see DOI 10.1099/ijsem.0.001448). You also do not say which strain of Ralstonia solanacearum you used (p. 13, lines 444–456). You need to add these strain names to the methods.
Response: Many thanks for your particular and constructive suggestions! In the revised manuscript, we added the “Rhizobial strain used here is Mesorhizobium loti MAFF303099. ” in Line 365. Similarity, we added the pathogen stain name as “Ralstonia solanacearum strain FQY_4” in Line 464 in the “Materials and Methods”.
- Figure Improvements Figure 2: The caption should include a brief explanation of the top row of images (control images). Figure 4: Figure 4A should be split into two figures –or– the two graphs need to be combined with a single y-axis. If you decide to split the two figures, then the text on pp. 5–6, lines 170–190 will need to be updated with the correct Figure numbers. The label of the y-axis of Figure 4B says "lenght" (should be "length"). Add labels to Figure 4D so that the reader knows what genotypes are being represented. Also, you need to indicate in the caption whether the error bars in Figures 4B, 4C, 4E, 4H, and 4K represent standard deviation or standard error of the mean. Figure 5: (D) shows the stainability chart and (E) is photographs of in vitro pollination assays. The caption has these mixed up. Also, you need to indicate whether the error bars in 4B, 4D, and 4F represent standard deviation or standard error of the mean. Figure 6: The caption for Figure 6A says "(label the x axis 'Days post-infection')". Instead of saying this in the caption, add it as a label. Also, it looks like you've mixed up part of the captions for Figures 6A, 6B, and 6C. 6A says that it represents "Numbers of colony forming units (CFU)…" but it is a growth curve. Figure 6B and Figure 6C says that they represent "Growth curves of R. solanacearum strains…" but they show numbers of CFUs. The caption for Figure 6C should indicate at what dpi the leaves were sampled. Finally, you need to indicate in the caption whether the error bars in Figures 6B, 6C, 6D, and 6E represent standard deviation or standard error of the mean. Figure 7: The images in Figure 7C have poor contrast, so it's hard to make out where the nodules are. It would help if you drew white triangles pointing at them. Also, you need to indicate in the caption whether the error bars in Figures 7A, 7B, and 7D represent standard deviation or standard error of the mean. Figure 8: The caption for Figure 8 should describe the error bars. Since you only have 3 replicates per condition, make sure these are standard deviations.
Response: Thank you so much for your careful review! In the caption of Figure 2, we added a sentence “The top row of images indicates the control.” in the revised manuscript. In Figure 4, we combined some graphs with a single y-axis and revised the caption accordingly. We corrected “Length” in Figure 4 C and marked genotype in Figure 4B. “Error bars indicate the SD of three biological replicates” was added in the caption. In Figure 5, we mixed the caption of 5D and 5E, now we revised the caption and also added standard deviation in 5 B, D, F. In Figure 6, we labeled the x axis 'Days post-infection' in Figure 6A, corrected the description of 6A, 6B and 6C, added “Detached leaves from 8-week-old plants were infected with R. solanacearum” in 6C and also showed “error bars in Figures 6B, 6C, 6D, and 6E represent standard deviation”. In Figure 7, we marked nodules on the roots in Figure 7C and added ± SD in the caption. In Figure 8, we described error bar in the caption.
Figure S1: It would be easier to see the Lj genes if you made them boldface or highlighted them. Figure S2: Since you have few replicates (3), standard error of the mean is inappropriate. You should use standard deviation, instead. Figure S4: The caption for Figure S4B should describe the error bars. Since you only have 3 replicates per condition, make sure these are standard deviations. Figure S5: This figure should be mentioned for the first time somewhere in section 2.4, not just in the discussion. Also, you need to indicate in the caption whether the error bars in Figure S5B represent standard deviation or standard error of the mean.
Response: Many thanks for the attentive comments! We added blue triangles in Figure S1 to indicate LjAGD homologs and described SD in Figure S4 and S5. in 2.4 we added a sentence “ In view of divergent roles of Arabidopsis AGD2 homologs in defense signaling [14], we further investigated the resistance phenotype of ljald1 mutants and the results showed that ljald1 mutants were more susceptible to R. solanacearum than wildtype plants (Figure S4)” to describe the resistance phenotype of ljald1.
- Terminology p. 4, line 134: E. coli cannot express genes that still have their introns. Since LjAGD2a and LjAGD2b have introns, you cloned the CDSs not the ORFs. Update this to reflect the correct terminology
Response: Many thanks for your carefully review and nice comments! We used CDSs to replace ORFs in the revised manuscript.
Reviewer 2 Report
Dear Authors,
In the manuscript of Huang et al. "Roles of AGD2a in plant development and microbial interactions of Lotus japonicus", a colossal work has been done to study the function of the AGD2 protein in the legume Lotus japonicus. It was shown that in L. japonicus there are 4 homologues of the gene encoding this protein, which has aminotransferase activity. However, the manuscript should be improved.
The title of the manuscript contains a section of the study - "microbial interactions", however, this topic is disclosed one-sidedly. The section of interaction with the pathogen is well presented, but symbiotic interactions are not well represented. So, in Figure 1, GUS staining in infected cells in the case of LjAGD2a is not noticeable, in the case of LjAGD2b, weak GUS staining was observed in a few infected cells (almost an empty nodule). It would be more useful to present a histological study, not to mention ultrastructural (this could be the subject of a separate article). But I consider the study of the phenotypic manifestation of nodule formation to be insufficient, especially since, according to the available data, nodule formation is ineffective (in the case of LjAGD2b).
The discussion section should also be expanded, in which the insights of the study of AGD2 proteins "in the microbial interactions of L. japonicus" was summarized. Only a few lines are devoted to discussing the participation of this protein in symbiotic interactions, where there is no mention of the effect of salicylic acid on nodulation and articles similar in subject matter are not discussed (for example: Stacey G., McAlvin C.B., Kim S.Y., Olivares J., Soto M.J. Effects of endogenous salicylic acid on nodulation in the model legumes Lotus japonicus and Medicago truncatula. Plant Physiol., 2006, 141(4): 1473-1481).
Author Response
In the manuscript of Huang et al. "Roles of AGD2a in plant development and microbial interactions of Lotus japonicus", a colossal work has been done to study the function of the AGD2 protein in the legume Lotus japonicus. It was shown that in L. japonicus there are 4 homologues of the gene encoding this protein, which has aminotransferase activity. However, the manuscript should be improved.
The title of the manuscript contains a section of the study - "microbial interactions", however, this topic is disclosed one-sidedly. The section of interaction with the pathogen is well presented, but symbiotic interactions are not well represented. So, in Figure 1, GUS staining in infected cells in the case of LjAGD2a is not noticeable, in the case of LjAGD2b, weak GUS staining was observed in a few infected cells (almost an empty nodule). It would be more useful to present a histological study, not to mention ultrastructural (this could be the subject of a separate article). But I consider the study of the phenotypic manifestation of nodule formation to be insufficient, especially since, according to the available data, nodule formation is ineffective (in the case of LjAGD2b).
Response: Thank you for nice suggestions! We used M.loti expressing DsRED to infect pLjAGD2a:GUS roots and bacteriods in the infection zone showed red florescence (Fig.1G), which clearly reflected that LjAGD2a was expressed mainly in the vascular bundle of nodules, not in the infected cells. Fig 1F, G, H, M have been replaced by new histological photographs and description of results was also modified in the revised manuscript from Line 90-94.
The discussion section should also be expanded, in which the insights of the study of AGD2 proteins "in the microbial interactions of L. japonicus" was summarized. Only a few lines are devoted to discussing the participation of this protein in symbiotic interactions, where there is no mention of the effect of salicylic acid on nodulation and articles similar in subject matter are not discussed (for example: Stacey G., McAlvin C.B., Kim S.Y., Olivares J., Soto M.J. Effects of endogenous salicylic acid on nodulation in the model legumes Lotus japonicus and Medicago truncatula. Plant Physiol., 2006, 141(4): 1473-1481).
Response: Thank you for the constructive suggestion! We added a sentence about the effect of endogenous SA on nodulation in Legumes. From 336-341, “Knockdown of LjALD1 decreased SA content to promote nodulation in L. japonicus [13]. Consistent with this result, overexpression of salicylate hydroxylase NahG in M. truncatula and L. japonicus resulted in the reduction of endogenous SA levels and therefore a greater number of infection threads and nodules [19]. ”. We also added the reference in the revised article accordingly.
Round 2
Reviewer 2 Report
Dear Authors,
The manuscript looks more presentable. You have changed Figure 1 well, now you can see that LjAGD2b nodules are not empty. It is difficult to judge how effective they are, since no test was made on the activity of nitrogen fixation (acetylene reduction) and the color of the nodules was not mentioned (pink - effective).
Minor remarks:
4.1. - there is no mention of the growing medium for Mesorhizobium loti, although it is carefully prescribed for Ralstonia solanacearum. The type or brand of the growth chamber is not mentioned, as well as the illumination intensity.
You describe how you observed pollen morphology, but you don't describe how you count the infection threads.
Author Response
The manuscript looks more presentable. You have changed Figure 1 well, now you can see that LjAGD2b nodules are not empty. It is difficult to judge how effective they are, since no test was made on the activity of nitrogen fixation (acetylene reduction) and the color of the nodules was not mentioned (pink - effective).
Response: Thank you so much for your nice comments. We showed the expression pattern of LjAGD2s in L. japonicus using pLjAGD2s:GUS plants in Figure 1, therefore, the nodule structure here cannot reflect whether LjAGD2s affect nitrogenase activity or not. But we always see pink nodules in both ljagd2 mutants and OE plants, so in this article we did not test the activity of nitrogen fixation.
Minor remarks:
4.1. - there is no mention of the growing medium for Mesorhizobium loti, although it is carefully prescribed for Ralstonia solanacearum. The type or brand of the growth chamber is not mentioned, as well as the illumination intensity.You describe how you observed pollen morphology, but you don't describe how you count the infection threads.
Response: Thank you so much for your nice suggestions. We add YM medium, the brand of the growth chamber, illumination intensity and also how to count infection threads in the revised manuscript from line 424-430.